# A Comprehensive Genetic Study of Microtubule-Associated Gene Clusters for Male Infertility in a Taiwanese Cohort

**DOI:** 10.3390/ijms242015363

**Published:** 2023-10-19

**Authors:** Chying-Chyuan Chan, Te-Hsin Yen, Hao-Chen Tseng, Brang Mai, Pin-Kuan Ho, Jian-Liang Chou, Gwo-Jang Wu, Yu-Chuan Huang

**Affiliations:** 1Graduate Institute of Medical Sciences, National Defense Medical Center, Taipei 114201, Taiwan; drobsgyn@seed.net.tw (C.-C.C.);; 2Department of Obstetrics and Gynecology, Taipei City Hospital-Renai Branch, Taipei 103212, Taiwan; 3School of Pharmacy, National Defense Medical Center, Taipei 114201, Taiwan; 4School of Dentistry, National Defense Medical Center, Taipei 114201, Taiwan; 5Department of Research and Development, National Defense Medical Center, Taipei 114201, Taiwan; 6Reproductive Medical Center, Department of Obstetrics and Gynecology, Tri-Service General Hospital, Taipei 114202, Taiwan

**Keywords:** male infertility, azoospermia, oligozoospermia, next-generation sequencing, single nucleotide polymorphism, microtubule-associated genes

## Abstract

Advanced reproductive technologies are utilized to identify the genetic mutations that lead to spermatogenic impairment, and allow informed genetic counseling to patients to prevent the transmission of genetic defects to offspring. The purpose of this study was to identify potential single nucleotide polymorphisms (SNPs) associated with male infertility. Genetic variants that may cause infertility are identified by combining the targeted next-generation sequencing (NGS) panel and whole exome sequencing (WES). The validation step of Sanger sequencing adds confidence to the identified variants. Our analysis revealed five distinct affected genes covering seven SNPs based on the targeted NGS panel and WES data: *SPATA16* (rs16846616, 1515442, 1515441), *CFTR* (rs213950), *KIF6* (rs2273063), *STPG2* (r2903150), and *DRC7* (rs3809611). Infertile men have a higher mutation rate than fertile men, especially those with azoospermia. These findings strongly support the hypothesis that the dysfunction of microtubule-related and spermatogenesis-specific genes contributes to idiopathic male infertility. The *SPATA16*, *CFTR*, *KIF6*, *STPG2*, and *DRC7* mutations are associated with male infertility, specifically azoospermia, and a further examination of this genetic function is required.

## 1. Introduction

Infertility is a reproductive disorder defined as the failure to achieve clinical pregnancy 12 months after unprotected intercourse [1,2]. The World Health Organization estimates that 9% of couples worldwide struggle with fertility problems, 50% of which are caused by male factors. Y chromosome microdeletion (YCMD) testing is a prevalent method for evaluating male infertility. The Y chromosome’s long arms (Yq) harbor three notable types of azoospermia factor regions: AZFa, AZFb, and AZFc. These regions, notably AZFc, are more prone to deletions, and the occurrence of such cases could disrupt spermatogenesis and testes development. While YCMD and the genetic underpinnings of male infertility have garnered significant research attention, the clinical implications within this realm remain scattered and lack a unified consensus.

Microtubules (MTs) are the structural components of the cells designed to participate in many essential processes such as cell division, intracellular transport, and cell shape maintenance. Similar to cells, sperms require MTs to allow proper assembly and retain proper functionality. In spermatogenic cells, MTs are responsible for the assembly of flagella in spermatids, and the production and maintenance of mature sperm motility [3]. 

Motor proteins in MTs are specialized molecular motors that use adenosine triphosphate (ATP) as an energy source. These motors are responsible for transporting various cellular cargoes along MTs tracks and ensuring the delivery of these cargoes to specific destinations within the cell; examples of such motor proteins include kinesins and dyneins [4]. Movement in the outward or anterograde direction (from sperm head to tail) is directly facilitated by the MT motor protein kinesin-2. In contrast, cytoplasmic dynein 2 (also known as dynein 1b) promotes movement in an inward or retrograde direction [5].

Furthermore, MTs ensure the successful division of cells during spermatogenesis [6]. As the sperm tail develops, the matrix built within the centrosome of the spermatid anchors the MTs. Within this region, MTs play a crucial role in facilitating the movement of vesicles from the Golgi to the acrosome [7]. MTs are also constituent elements of the sperm tail, the elongated flagella whose central axoneme protrudes from the basal body located posterior to the nucleus. To enable sperm motility, the movement of the inner and outer duplex MTs of the sperm flagella is dependent on the energy generated by the hydrolysis of ATP [8].

The Katanin catalytic subunit A1 like 1 (*KATNAL1*) gene is a protein-coding gene that is expressed in testicular tissue and helps manage Sertoli cell microtubules (MTs) dynamics. Katanin p80 is critical for regulating MTs and is a vital component for understanding the molecular mechanisms of male infertility. The dysfunction of the Katanin p80 subunit results in low sperm counts, poor motility, and abnormal sperm morphology [9]. Studies in mutant mice [10] indicate that loss-of-function mutations in *KATNAL1* disrupt MT function and can cause male infertility by affecting spermatogenesis. Subsequent human studies [11] also supported that mutations in the *KATNAL1* gene may facilitate male infertility. Furthermore, Wu et al. [12] also highlighted the importance of MTs-related gene defects (particularly *KIF15* and dynamin 1) in azoospermia samples by using single-cell transcriptome analysis (scRNAseq). Mutations of the N-terminal and C-terminal of *DNAH1* can have different effects on the axoneme structure of human spermatozoa [13]. Genetic mutations that disrupt MTs’ function during sperm development may cause male infertility. More studies also have shown that the regulation of MT dynamics is critical for male infertility (azoospermia and oligospermia) [14,15].

This study utilized the targeted NGS panel and WES to explore male infertility (MI)-associated SNPs to understand their causes and search for human azoospermia and oligospermia gene clusters that lead to abnormal sperm function. The aim of this study was to identify MTs-related genes (such as *KIF6* and *DRC7*) and their properties in sperm cell function, motility, intracellular trafficking, differentiation, and cell division. The assessment of MTs’ dynamics is an important aspect to fully understand the correlation between genes and male infertility. Further research in this area may improve diagnosis, treatment, and interventions for male infertility.

## 2. Results

### 2.1. Semen Analysis

The mean age of the fertile controls, oligozoospermic, and azoospermic patients were 37, 38, and 37 years old, respectively. The semen volumes in oligozoospermia and azoospermia were abnormal, especially in azoospermia (1.64 ± 1.8, *n* = 15, *p* < 0.05). 

### 2.2. Y Chromosome Microdeletion

Y chromosome microdeletions (YCMD) are identified in approximately 13% of men with nonobstructive azoospermia and approximately 5% of men experiencing severe oligozoospermia. Within the framework of this study, YCMD analysis (Appendix A) was carried out on a group of eight infertile men, comprising four with oligozoospermia and four with azoospermia. Interestingly, notable AZFc deletions were observed solely in two azoospermic patients (S02 and S-2017). The microdeletion of AZF-STS was not identified in two of the infertile patients, while others exhibited certain degrees of microdeletion (Appendix A, Appendix A). The extent was insufficient to imply a direct cause-and-effect relationship.

### 2.3. Variant’s Analysis and Validation in Targeted NGS

Targeted sequencing was performed on eight infertile men and two fertile controls. SNPs were identified from the 15 spermatogenesis-related genes (Table 1). The primers of the *SPATA16*, *CFTR,* and *ESR1* genes were used for the Sanger sequencing validation of NGS variants, and the sequences are shown in Appendix A. *SPATA16* mutations at three SNPs (rs16846616), (rs1515442), and (rs1515441) were identified in all azoospermic patients and one oligozoospermic patient, while no mutation was displayed in fertile patients (Table 2). The *CFTR* mutation at one SNP (rs213950) was identified in seven of eight infertile men and one of two fertile controls, though when verified with Sanger sequencing, no fertile controls displayed this *CFTR* mutation (Table 2). The remaining genes seemed to correlate more poorly. The mutations at the SNPs of *TEX11* (rs4844247), *LHB* (rs4146251380), *USP26* (rs61741870) and (rs41299088), and *ANOS1* (rs2229013) were displayed in only one of eight infertile men. *NR5A1* (rs1110061) and *ANOS1* (rs808119) mutations were present in both fertile controls. *TEX11* (rs6525433) displayed mutations in only two of eight infertile men. GNRH1 displayed mutations in five of eight infertile men and one of two fertile controls, however this was not validated with Sanger sequencing (Table 2). *ESR1* displayed mutations in five of eight infertile men and none in fertile controls (Table 2), and could be further explored as a gene target for male infertility. The results of the Sanger sequencing examining the *SPATA16* and *CFTR* genes are presented in Table 3 and Figure 1. No fertile controls displayed the *CFTR* mutation while 6 of 16 oligozoospermic and 10 of 15 azoospermic patients did, suggesting that *CFTR* may be associated with male infertility (oligozoospermia and azoospermia). The mutations of *SPATA16* at two SNPs (rs16846616) and (rs1515411) occurred simultaneously, affecting 12 of 15 azoospermic, 6 of 16 oligozoospermic, and 2 of 8 fertile control patients. The mutation of *SPATA16* at one SNP (rs1515442) affected 9 of 15 azoospermic, 1 of 16 oligozoospermic, and none of the fertile control patients. The *SPATA16* mutations appear to be associated with azoospermia. Three nearby SNPs (rs1515442), (rs1515441), and (rs16846616) in *SPATA16* were shown to have a 4~9.6-fold higher mutation incidence in azoospermia compared to oligozoospermia, while *CFTR* was only 1.8-fold higher (Figure 1). 

Using CADD, SIFT, and PolyPhen2, we performed an in silico evaluation of the predicted pathogenicity of the *SPATA16* and *CFTR* mutations. In terms of CADD, the *SPATA16* SNP was identified as a deleterious variant. Their values were 15.67, 19.12, and 22.2, which surpassed the deleterious variant cutoff of 15 (Appendix A). They are also deleterious in SIFT. Only one nucleotide variant of *SPATA16* (rs1515442) appeared benign in PolyPhen2. Overall, *SPATA16* mutations at two SNPs (rs16846616 and rs1515441) are expected to be deleterious. MAF was examined in gnomAD and TWB, and the results are shown in Appendix A. Two SNPs (rs16846616) and (rs1515411) of *SPATA16* had lower MAF frequencies of 13.1% and 11.9%, respectively, in genomAD compared with TWB (32.8 and 32.5%). Interestingly, in our study in Taiwan, *SPATA16* SNPs (rs16846616) and (rs1515441) co-occur at an 80% higher frequency in azoospermic patients (Appendix A).

### 2.4. SNPs of Microtubule-Associated Genes in WES

Based on the variant comparison data (*p* < 0.05) from CLC cases and controls, we have provided a detailed list of 15 SNPS in 13 candidate genes with significant expression in testicular tissues, including *KIF6*, *STPG2*, *DRC7*, *NEK2*, *TRIM49*, *CATSPER2*, *CMTM2*, *SART3*, *DYNC2H1*, *CCDC168*, *BORCS5*, *TPTE*, and *RADIL* (Appendix A). Among these, certain genes exhibit a high global Minor Allele Frequency (MAF) (>0.3) and a higher mutation rate in controls (>0.3). The genes with a high MAF and mutation rate in controls include *TRIM49*, *CCDC168*, *TPTE*, and *RADIL*, which might lead to an apparent elevated mutation rate in controls as well. The mutation rates and MAF for these genes in both cases and controls are presented in Appendix A. 

### 2.5. Variant Analysis and Validation in WES

We selected two candidate genes, *KIF6* and *STPG2*, from the pool of 60 candidates for Sanger sequencing validation. Sanger sequencing was employed to confirm the presence of SNPs on *KIF6* and *STPG2*, facilitating a gene comparison between infertile and fertile men (Table 4). In the case of *KIF6*, 1 out of 8 controls exhibited mutation, whereas 22 out of 31 cases harbored mutations. Similarly, for *STPG2*, there were 2 mutations among the 8 controls and 20 mutations among the 31 cases. Furthermore, in the *KIF6* gene, both the A/A and G/A types of SNPs were observed (Appendix A). Upon comparing infertile men to fertile men (fertile within 3 years), the occurrence of SNPs on the *KIF6* gene appeared to be notably higher by a multiple of 5 (as shown in Figure 2). 

Using CADD, SIFT, and PolyPhen2, we conducted an *in silico* evaluation of the predicted pathogenicity of 13 mutations. In terms of CADD scores, five SNPs surpassed the deleterious variant cutoff of 15 (Appendix A), namely *STPG2*, *NEK2*, *DYNC2H1*, *BORCS5*, and *TPTE*. Among them, *BORCS5* (CADD score: 29.9) and *STPG2* (CADD score: 25.6) exhibited the highest scores. When considering SIFT predictions, *BORCS5*, *STPG2*, *DRC7*, and *CATSPER2* were classified as deleterious variants. Within the framework of PolyPhen2 analysis, *STPG2*, *TRIM49*, *BORCS5*, and *TPTE* were designated as possibly damaging, suggesting that alterations in these amino acids might disrupt protein structure. Minor Allele Frequency (MAF) was assessed both globally and specifically in East Asian populations (MAF_eas), with the results also summarized in Appendix A. Notably, *KIF6* (MAF: 0.04), *BORCS5* (MAF: 0.06), *DRC7* (MAF: 0.24), and *STPG2* (MAF: 0.39) were listed in the global MAF. It is worth highlighting that *STPG2* exhibited a nearly 40% mutation rate, despite demonstrating a favorable case-to-control ratio (88/14) in whole exome sequencing (WES) data. However, this high mutation frequency in *STPG2* could impact its potential as a candidate diagnostic marker. Additionally, it is noteworthy that *KIF6* displayed significant differences in both global MAF (4%) and MAF_eas (25%), indicating a higher mutation rate in East Asian male populations. Interestingly, in our Taiwan study, the *KIF6* SNP (rs2273063) and *STPG2* (rs2903150) co-occur in an 80% mutation rate in azoospermia and oligozoospermia patients with 20% MAF in ordinary men of Taiwan by Sanger sequencing.

## 3. Discussion

Genetic screening to identify the genetic mechanisms of spermatogenesis failure in infertile men has become of clinical importance. These results not only allow us to determine the etiology but also prevent the iatrogenic transmission of genetic defects to offspring through assisted reproductive techniques. These goals pose enormous challenges to reproductive medicine. In this study, we found some MI-related genes associated with male infertility. Through targeted NGS, we extensively examined 15 candidate genes involved in spermatogenesis (257 amplicons, 27,707 bp) within a cohort of unrelated Taiwanese infertile men (*n* = 8) (see Table 1). Building upon previous studies, we noted the critical roles of *SPATA16* in globozoospermia and *CFTR* in obstructive oligozoospermia or azoospermia [16]. Notably, our findings provide further support for mutations in *SPATA16* and *CFTR* genes being associated with non-obstructive oligozoospermia and azoospermia with normal morphology. Intriguingly, we noted a strong correlation between *SPATA16* and azoospermia in our patient group. Additionally, our study raises questions about the implications of *ESR1* and *GnRH1* gene mutations in male infertility, which warrant a more in-depth investigation. When discussing MAF, we should recognize that variant allele frequency is basically a fraction, and the variant positivity rate divided by the total number of alleles is screened.

### 3.1. Microtubule-Associated Genes Affect Spermatogenesis by Variants Analysis

The ***KIF6*** (kinesin family member 6) gene encodes motor proteins, including kinesins and dyneins, that play crucial roles in intracellular transport and cellular movement. Their activity is particularly important in cellular divisions during spermatogenesis and sperm motility [17,18]. The two C-terminal tail domains interact with transported cargo through adapters and are connected to the head by a filamentous coiled stem that oligomerizes and regulates the dynamics of MTs [19,20,21,22]. A single nucleotide polymorphism (SNP, rs2273063) in the *KIF6* gene was identified through Illumina TruSeq whole exome sequencing (WES). This SNP had a higher prevalence in cases (87.5%) compared to controls (0%). Consequently, we validated SNPs by Sanger sequencing, which demonstrated that mutations in infertile cases were five times higher (0.65; 13/20) than in fertile controls (0.13; 1/8). Additionally, we performed a random sampling in ordinary men to estimate the mutation frequency of typical men in Taiwan (0.2; 4/20), contrasting it with the Minor Allele Frequency (MAF) (0.04; Table 4). This reveals that the incidence of *KIF6* SNPs collected from infertile men in Taiwan is 16.25 times higher than the global MAF, and 3.25 times higher than the ordinary men in Taiwan. Based on Sanger sequencing data, the frequencies observed in our cases and controls suggest that *KIF6* mutations account for 60% of azoospermia cases and 50% of oligospermia cases, respectively. This underscores a greater occurrence of infertility among men (with azoospermia and oligospermia) compared to fertile men (Table 4). Notably, its variant SNPs (A/G and A/A) may exhibit the potential for genetic diagnosis and serve as markers for the progression of spermatogenesis (Appendix A).

***DRC7*** (dynein regulatory complex subunit 7) is predicted to play a role in flagellated sperm motility, which aligns with its involvement in regulating ciliary motility [23,24]. Alongside flagellated sperm motility, it has potential involvement in spermatogenesis and the development of sperm cells. A single nucleotide polymorphism (SNP, rs3809611) in the *DRC7* gene was identified through Illumina TruSeq whole exome sequencing (WES). The mutation rate of the *DRC7* gene is reported to be 75%, which is lower than the mutation rate of the *KIF6* gene we mentioned earlier. The *DRC7* and *KIF6* genes were the only two genes within the control group that exhibited no mutations. The *DRC7* gene has the lowest global Minor Allele Frequency (MAF) among all candidate genes, with MAF values of 0.24 (global) and 0.16 (East Asia). This indicates that the SNP is relatively uncommon in these populations. Both *DRC7* and *KIF6* stand out as promising candidates for meaningful genetic testing in the context of male infertility.

***STPG2*** (sperm tail PG-rich repeat containing 2) contains a PG-rich motif characterized by a five-residue pattern: P-G-P-x-Y, and forms a similar structure bound to the outer junction of MT [25]. The expression profile of the STPG2 protein suggests that it might be important in both testicular development and spermatogenesis and its deletion could impair spermatogenesis [25]. A single nucleotide polymorphism (SNP, rs3809611) in the *STPG2* gene was identified through Illumina TruSeq whole exome sequencing (WES). The frequency of mutations among infertile patients was 87.5% (7/8), while the control group with recent fertility exhibited a mutation frequency of 14.3% (1/7). To validate the SNP by Sanger sequencing, the ratio of *STPG2*-mutated cases (20/31) to controls (2/8) is approximately only 2.6. Furthermore, it is noteworthy that our random sampling in ordinary men displayed a similar mutation frequency of 40%, which matches with the MAF. Of particular interest, the Azoospermia subgroup showed a mutation rate of 80% (12/15), as did the oligospermia group with 50% mutations (8/16). However, the *STPG2* gene itself presents an almost 40% mutation rate globally, whereas in East Asian populations, this rate drops to 30%. Consequently, when evaluating its potential relevance to male infertility, it is less representative compared to *KIF6* due to its already elevated mutation rate in ordinary men.

***NEK2*** (NIMA-related kinase 2) is involved in centrosome duplication, a key process that ensures the correct organization of the MT organizing center of the cell [26]. Accurate cell division is essential for proper sperm development. Any disruption caused by abnormal *NEK2* activity could result in damaged sperm cells. These disturbances may lead to diseases such as male infertility [27]. *NEK2* has a global MAF of 0.16 and an East Asian MAF of 0.22. Its relatively low mutation rate can be used as a strong reference index for male infertility. A single nucleotide polymorphism (SNP, rs2230489) in the *NEK2* gene was identified by Illumina TruSeq whole-exome sequencing (WES), and the mutation rate of the cases was 75% (6/8), while the mutation rate of the control group was 14.3% (1/7).

### 3.2. Other Genes Affect Spermatogenesis by Variants Analysis

***SPATA16*** (spermatogenesis-related 16, also known as NYD-SP12) is highly expressed in human testis and localized to the Golgi apparatus and pro-acrosomal vesicles [28], which fuse to form the acrosome during spermatogenesis [29,30]. It was identified as the first autosomal gene and demonstrated that a homozygous mutation (c.848G→A) led to globozoospermia, the production of round-headed and acrosomeless spermatozoa. In this study, we report that three closely homozygous mutations in *SPATA16* (rs16846616), (rs1515442), and (rs1515441) are not only associated with male infertility, but especially azoospermia in our Taiwan cohort. In particular, *SPATA16* (rs16846616) and (rs1515441) co-occur with a higher frequency (80%) in our azoospermic patients of Taiwan.

***CFTR*** (cystic fibrosis transmembrane conductance regulator) is one of the most common genetic mutations leading to azoospermia, which can lead to abnormalities in the male reproductive tract and ultimately result in infertility [31,32]. In our Sanger sequencing study, we found that 51.6% (16 out of 31) of infertile men had *CFTR* mutations. These mutations were absent in ordinary men. Among the infertile men with *CFTR* mutations, 67% of those with azoospermia (lack of sperm in semen) and 37.5% of those with oligospermia (low sperm count) had these mutations. Based on our findings, we establish a strong association between mutations in the *CFTR* gene, specifically the rs213950 mutation, and the occurrence of oligospermia and azoospermia in this cohort of Taiwanese patients.

***CMTM2*** (CKLF-Like Marvel Transmembrane Domain Containing 2) plays a critical role in spermiogenesis in mice, which has been studied extensively [33]. *CMTM2* has been extensively studied and was found to play a crucial role in spermiogenesis in mice. This gene’s significance is particularly pronounced during the essential stages of sperm development. *CMTM2*−/− mice were unable to produce sperm, while *CMTM2*+/− mice exhibited a significant reduction in sperm count and motility. A single nucleotide polymorphism (SNP, rs2290182) in the *CMTM2* gene was identified by Illumina TruSeq whole-exome sequencing (WES), and the mutation rate of the cases was 75% (6/8), while the mutation rate of the control group was 14.3% (1/7). The Global Minor Allele Frequency (GMAF) and East Asian Minor Allele Frequency (MAF) were determined to be 0.17 and 0.28, respectively, indicating a higher prevalence within the East Asian population.

In this study, an Ampiseq targeted NGS panel and Illumina TruSeq WES were both used to pinpoint six specific genes (*KIF6*, *DRC7*, *STPG2*, *SPATA16*, *CFTR*, *CMTM2*) that play a role in MT association and spermiogenesis, which are crucial factors in understanding the causes of male infertility.

## 4. Materials and Methods

### 4.1. Patients and Controls

From July 2018 to January 2020, 31 infertile men between 30 and 45 years old were recruited during routine infertility treatment at the Reproductive Medical Center, Tri-Service General Hospital and Taipei City Hospital-Renai Branch (Taipei, Taiwan). Sperm concentration, motility function, and morphology are strongly associated with the genes of an individual. The genetic insights gained from sperm concentration studies may lead to the development of genetic tests or panels for assessing the risk probability for male infertility. Infertile men alongside their semen samples were analyzed (Table 5), and we divided them into two groups based on the sperm concentration in semen: azoospermia (<0.1 million/mL, *n* = 15) and oligozoospermia (0.1–15 million/mL, *n* = 16). The fertile controls (*n* = 9) were men who had children in the previous 3 years. The study protocol was approved by the Institutional Review Board of the Taipei City Hospital Research Ethics Committee (protocol Ver2.0-1090414) (Taipei, Taiwan), and all the patients provided written consent prior to enrollment in the study. 

### 4.2. Semen Analysis

Patients refrained from sexual activity for 3 to 5 days, then semen samples were collected in sterile containers by masturbation. The semen is kept at room temperature for 15 to 30 min to allow liquification. After the process of liquefaction, the total volume of the semen and the sperm concentration, motility, and shape were documented in accordance with the World Health Organization Laboratory Manual for the Examination and Processing of Human Semen, Fifth Edition. A semen volume less than 1.5 mL, sperm concentration less than 15 million/mL, motility less than 40%, progressive motility (PR) plus non-progressive motility (NP) less than 32%, or normal morphology in less than 30% were considered abnormal. After macroscopic and microscopic observations on sperm parameters, only sperm concentration is an evident factor. Patients were divided into three groups based on sperm concentration, fertile controls (≥15 million/mL), oligozoospermia (≤15 million/mL), and azoospermia (<0.1 million/mL). The sperm of infertile men and fertile controls were examined and summarized in Table 5. There were 16 patients categorized into the oligozoospermia group, 15 patients categorized into the azoospermia group, and 9 fertile controls.

### 4.3. Y Chromosome Microdeletion (YCMD) Examination

The Y chromosome’s long arms (Yq) harbor numerous coding genes responsible for regulating spermatogenesis and testes development. Microdeletions within the AZF (azoospermia factor) region can lead to a diverse range of infertility phenotypes. To investigate these deletions, Yq microdeletion analysis was conducted by amplifying the AZFa, AZFb, and AZFc loci along with their associated sequence-tagged sites (STSs) markers. By employing the YCMD assay (Promega, Madison, WI, USA), a PCR-based blood test, the presence or absence of sequence-tagged sites (STSs) became assessable alongside clinically relevant microdeletions (Appendix A). 

### 4.4. Targeted Next-Generation Sequencing (NGS) Panel

The amplicon libraries of eight infertile men and two fertile controls were constructed using the Ion AmpliSeq™ Library Kit v2.0 (ThermoFisher Scientific, Waltham, MA, USA) according to the manufacturer’s instructions. The Ion Xpress™ Barcode Adapter Kit (ThermoFisher Scientific, Waltham, MA, USA) was used for barcode adapter ligation. The patient’s genomic DNA was extracted from peripheral blood leukocytes using the MagPurix DNA extraction kit (Taipei, Taiwan) according to the manufacturer’s instructions. The custom Ion Ampliseq NGS panel was designed with 257 amplicons (IAD142966_182, 131 + 126 primer pairs containing 27,707 bps) to ensure a comprehensive coverage of 15 gene targets previously associated with male infertility (Table 1). Massively parallel sequencing was performed in the GeneStudio S5 Sequencing System (ThermoFisher Scientific, Waltham, MA, USA) to cover the exonic regions of genes, increasing the average sequencing depth by more than 1000-fold. All reads were further analyzed by Torrent Suite™ software 5.4 (ThermoFisher Scientific, Waltham, MA, USA) with the human reference genome (GRCh37.p5/hg19). All synonymous and non-altered protein splice site variants were then removed, leaving only the coding gene for comparison. In this study, spermatogenesis-related genes were used to design a male infertility targeted NGS panel which included *SPATA16, AURKC, CFTR, ESR1, TEX11, LHB, USP26, AR, GNRH1, NR5A1, KAL2, DAZL, PICK1, FSHB,* and *ANOS1*. Their coverage and the number of amplicons per gene in the targeted NGS panel are shown in Table 1. Ten of the fifteen genes have 100% coverage, and the designed amplicons of *SPATA16* and *CFTR* are shown in Appendix A while the other genes are shown in Appendix A–D.

### 4.5. Whole-Exome Sequencing (WES) and Variant Analysis

WES is a widely used next-generation sequencing (NGS) method that involves sequencing the protein-coding regions of the genome. We selected 15 samples of participants (8 cases and 7 controls) extracted from peripheral blood leukocytes by using the MagPurix DNA extraction kit, processed according to the TruSeq DNA Exome Kit (Illumina, San Diego, CA, USA) guidelines, and fragmented 100 ng of genomic DNA into 150 bp inserts by the M220 Focused-ultrasonicator (Covaris, Woburn, MA, USA). Briefly, fragment gDNA is ligated with adapters and enriched by PCR. After PCR amplification, the probe captured and amplified fragments to create the WES library. The Nextseq 550 (Illumina) was used for WES. Then, those sequenced reads were aligned to the human reference genome (hg38) by using NGS Core Tools/FASTQ mapping with CLC Genome Workbench 23.0.4 (Qiagen, Hilden, Germany). We filtered out and compared those detected variants with the SNPs database (*p* < 0.05). SNPs profiles were assessed by using the resequencing analysis/variant comparison module of CLCs. The atlas of NCBI HPA RNAseq was utilized to compare infertile men (experimental data) to fertile men (atlas), and our team noticed specific genes that were shown to have a significant expression in testicular tissues in infertile men.

### 4.6. Validation by Sanger Sequencing

After the genes of interest were identified using targeted NGS sequencing and WES, Sanger sequencing was applied to examine the potential SNPs of the *CFTR*, *SPATA16*, *KIF6*, *STPG2*, and *DRC7* genes. The product of targeted genes was gained by using Proflex PCR (ThermoFisher Scientific, Waltham, MA, USA) with a pair of primers, and the sequences of primers are listed in Appendix A. The Sanger sequencing of the Applied Biosystems 3130X Genetic analyzer (ThermoFisher Scientific, Waltham, MA, USA) was performed on the infertile cases and fertile controls to validate the findings of the candidate SNPs by WES and targeted NGS. Sanger sequencing has also been used to validate allelic variation in other larger groups of infertile men and fertile controls.

### 4.7. In Silico Evaluation Workflow

The in silico evaluation of the pathogenicity of nucleotide changes in exons was performed using CADD (Combined Annotation Dependent Depletion), SIFT (Sorting Intolerant from Tolerant), and PolyPhen2 (Polymorphism Phenotyping) (applied in Appendix A). CADD is a tool for scoring the deleteriousness of single-nucleotide variants in the human genome, with a cutoff score of 15 with higher values indicating more deleterious conditions. The PolyPhen-2 score predicts the likely impact of amino acid substitutions on human protein structure and function, with scores ranging from 0.0 (tolerated) to 1.0 (deleterious). SIFT predicts whether amino acid substitutions are likely to affect protein function based on scores and qualitative predictions (either ‘tolerated’ or ‘deleterious’). Minor Allele Frequencies (MAFs) were examined in the genome aggregation database gnomAD and Taiwan Biobank (TWB).

## 5. Conclusions

Male infertility-associated genes can be roughly divided into three affected clusters: tubulin-associated genes (dynein and kinesin), outer junction proteins-associated genes, and spermatogenesis-associated genes. Our experimental results have pinpointed evidence for a potential link between genetic variations in the *KIF6*, *DRC7*, and *STPG2* genes with male infertility (Figure 3). The fact that the prevalence of these SNPs was found to be significantly higher in infertile men compared to fertile men suggests that these genetic variations could indeed be associated with the development of male infertility. The alignment of our findings with the significance of MT-related genes, particularly in the context of sperm cell structure and function, adds an additional layer of biological plausibility to the observed associations. MTs are essential components of the cytoskeleton and play crucial roles in cellular processes, including the formation of sperm heads and tails. The potential for a genetic diagnosis based on these findings is promising, as it suggests that certain genetic variations could serve as markers for identifying individuals at a higher risk of male infertility. Our further approach also includes CRISPR gene knockout experiments to gain deeper insights into the precise roles of these candidates and their potential associations with male infertility. By involving larger azoospermic and oligozoospermic cohorts, we anticipate accelerating the identification of novel genes contributing to these phenotypes, ultimately contributing to a more comprehensive understanding of male infertility.

## Figures and Tables

**Figure 1 ijms-24-15363-f001:**
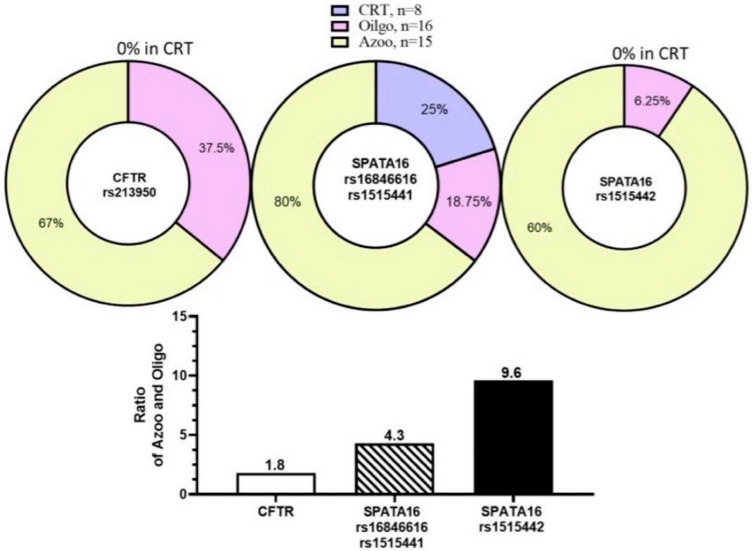
Three nearby SNPs (rs1515442, rs1515441, and rs16846616) in *SPATA16* showed a 4~9.6-fold mutation incidence in azoospermia compared to oligozoospermia, while *CFTR* was only 1.8-fold higher.

**Figure 2 ijms-24-15363-f002:**
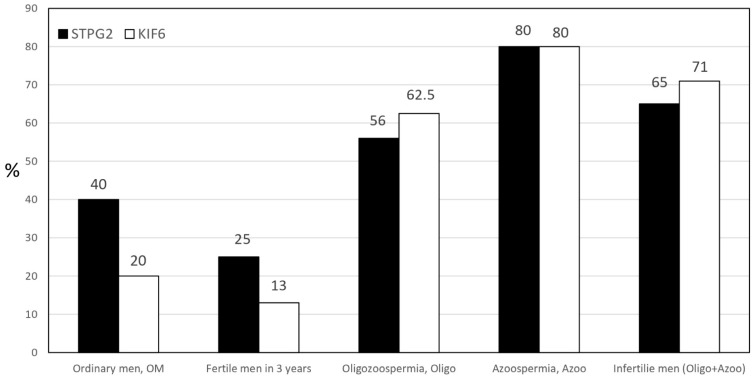
The incidence of SNPs on *KIF6* gene in infertile men (azoospermia) was more than six times higher compared to fertile men by Sanger sequencing.

**Figure 3 ijms-24-15363-f003:**
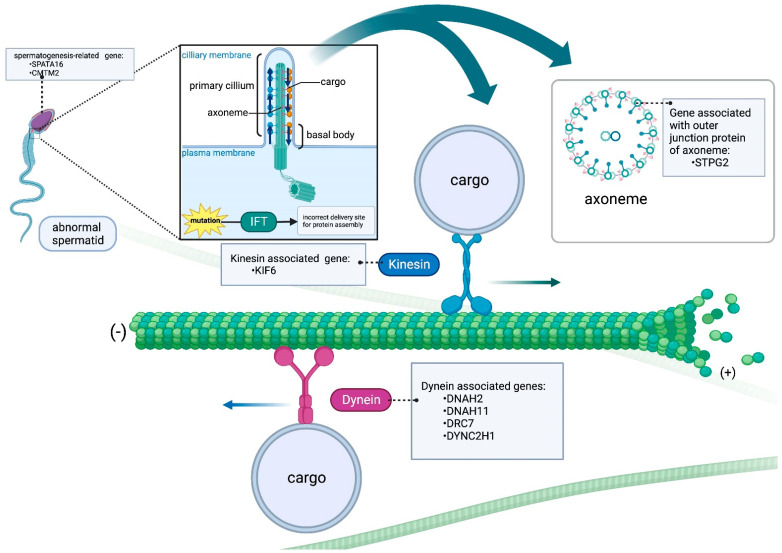
Male infertility-associated genes discovered in this study can be roughly divided into three affected clusters: MT-associated genes (dynein and kinesin), outer junction proteins-associated genes, and spermatogenesis-associated genes.

**Table 1 ijms-24-15363-t001:** The customized Ion Ampliseq NGS panel of infertility included 15 spermatogenesis-related genes with 257 amplicons (IAD142966_182, 131 + 126 primers). We performed massive parallel sequencing on an Ion S5 semiconductor sequencer (ThermoFisher Scientific, Waltham, MA, USA)).

Gene ID	Gene Name	Chromosome	Number of Amplicons	Total_Bp	Covered_Bp	Missed_Bp	Coverage
G1	*FSHB*	chr11	3	410	410	0	1
G2	*AURKC*	chr19	9	1011	1011	0	1
G3	*LHB*	chr19	3	456	191	265	0.419
G4	*PICK1*	chr22	17	1368	1368	0	1
G5	*SPATA16*	chr3	16	1810	1810	0	1
G6	*DAZL*	chr3	15	1071	1071	0	1
G7	*ESR1*	chr6	15	1874	1874	0	1
G8	*CFTR*	chr7	46	4713	4709	4	0.999
G9	*KAL2*	chr8	28	2825	2825	0	1
G10	*GNRH1*	chr8	4	321	321	0	1
G11	*NR5A1*	chr9	14	1446	1424	22	0.985
G12	*AR*	chrx	19	2873	2873	0	1
G13	*TEX11*	chrx	34	3160	3068	92	0.971
G14	*USP26*	chrx	15	2752	2752	0	1
G15	*KAL1 (ANOS1)*	chrx	19	2183	2000	183	0.916

**Table 2 ijms-24-15363-t002:** The point mutation of genes in infertile men shown in red square (■); the point mutation of genes in fertile men shown in black square (■). (Azoo: azoospermia; Oligo: oligozoospermia).

Gene	Chr.	SNP ID	Allele Variation	Oligo				Azoo				Fertile Men	
				S06	S07	S08	S11	S02	S09	S10	S-2017	N1	CRT-1
*SPATA16*	3	rs16846616	T→C			■		■	■	■	■		
		rs1515442	C→T			■		■	■	■	■		
		rs1515441	C→T			■		■	■	■	■		
*CFTR*	7	rs213950	G→A	■	■	■	■		■	■	■		■
*ESR1*	6	rs17847065	C→A	■	■		■		■	■			
*TEX11*	X	rs6525433	T→C	■			■						
		rs4844247	C→T				■						
*LHB*	19	rs146251380	G→A,T								■		
*USP26*	X	rs61741870	A→G					■					
		rs41299088	G→A					■					
*AR*	X	rs777131133	C→A,G,T					■				■	
*GNRH1*	8	rs6185	C→A,G,T		■	■	■			■	■	■	
*NR5A1*	9	rs1110061	C→A,G	■	■		■			■		■	■
*ANOS1*	X	rs808119	C→A,T	■	■	■						■	■
		rs2229013	C→A,T					■					

**Table 3 ijms-24-15363-t003:** The incidence of rs1515442, rs1515441, and rs16846616 on the *SPATA*16 gene by using Sanger sequencing.

Phenotype	Patient ID	*CFTR*	*SPATA16*		
		rs213950	rs16846616	rs1515442	rs1515441
		G→A	T→C	C→T	C→T
	CRT-1				
	N1				
	N2				
CRT, Fertile	N3		T→C		G→A
(*n* = 9)	N4		T→C		G→A
	N5				
	N8				
	N27(NN7)				
	N31(sample 1)				
	S01				
	S03	C→T	T→C		G→A
	S04				
	S05				
	S06	C→T			
	S07	C→T			
	S08	C→T	T→C	C→T	G→A
Oligo, sperm counts	S11	C→T			
≦15 million (*n* = 16)	S15				
	S18				
	S20				
	CA21				
	CA22	C→T	T→C		G→A
	CA24				
	CA25				
	S31				
	P1 (S-2017)	C→T	T→C	C→T	G→A
	P2		T→C	C→T	G→A
	S02	C→T	T→C	C→T	G→A
	S09	C→T	T→C	C→T	G→A
	S10	C→T	T→C	C→T	G→A
	S13		T→C		G→A
Azoo, sperm counts	S14				
<0.1 million (*n* = 15)	S16		T→C		G→A
	S17	C→T			
	S26	C→T	T→C	C→T	G→A
	S32	C→T	T→C	C→T	G→A
	CA23		T→C		G→A
	CA26	C→T			
	TCA26	C→T	T→C	C→T	G→A
	TS1	C→T	T→C	C→T	G→A

**Table 4 ijms-24-15363-t004:** The occurrence of mutations on the *STPG2* and *KIF6* genes was validated through Sanger sequencing. Thirty-one (31) infertile men with no or low semen sperm count (≤15 million/mL), twenty ordinary men, and eight fertile men.

Phenotype	Patient	*STPG2*	*KIF6*
	N1	A/G	X
	N2	X	G/A
	N3	A/G	G/A
	N4	X	X
	N5	X	X
	N6	A/G	X
	N7	X	X
	N8	X	X
	N9	A/G	X
Ordinary Men	N10	A/G	G/A
*n* = 20	N11	X	X
	N12	X	G/A
	N13	A/G	X
	N14	X	X
	N15	X	X
	N16	X	X
	N17	X	X
	N18	A/G	X
	N19	X	X
	N20	A/G	X
	PN1	X	X
	PN2	X	G/A
	PN3	X	X
CRT Fertile	PN4	X	X
*n* = 8	PN5	A/G	X
	PN6	X	X
	PN7	A/G	X
	NN7	X	X
	S01	A/G	G/A
	S03	A/G	G/A
	S04	A/G	G/A
	S05	X	X
	S06	X	G/A
	S7	A/G	G/A
Oligozoospermia	S8	A/G	A/A
*n* = 16	S11	X	G/A
	S12	A/G	X
	S15	A/G	G/A
	S18	X	G/A
	S20	X	X
	CA22	X	X
	CA24	X	X
	CA25	A/G	A/A
	S31	X	X
	P1	A/G	G/A
	P2	A/G	G/A
	S02	A/G	G/A
	S09	A/G	G/A
	S10	A/G	G/A
	S13	A/G	G/A
Azoospermia	S14	A/G	G/A
*n* = 15	S16	A/G	G/A
	S17	X	X
	S26	X	X
	S32	A/G	G/A
	CA23	A/G	X
	CA26	A/G	G/A
	TCA26	X	G/A
	TS1	A/G	G/A

**Table 5 ijms-24-15363-t005:** Thirty-one (31) infertile men with no (Azoo, *n* = 15) or low (Oligo, *n* = 16) semen sperm count (≤15 million/mL, oligo) and nine fertile men (CRT) as controls were included in this study (nd: not detectable).

Group by	Patient	Volume	Sperm Counts	Age	Motility	Morphology
Phenotype	ID	(mL)	(M)		(%)	(%)
	CRT-1	3.2	normal	40	45	60
CRT Fertile (*n* = 9)	N1	2.1	normal	32	35	36
	N2	nd	normal	33	nd	nd
	N3	nd	normal	35	nd	nd
	N4	nd	normal	42	nd	nd
	N5	nd	normal	38	nd	nd
	N8	nd	normal	29	nd	nd
	N27(NN7)	nd	normal	40	nd	nd
	N31	nd	normal	50	61	61
Oligo, sperm counts ≤ 15 million (*n* = 16)	S01	1.9	15	42	13	58
	S03	1.8	11.6	40	65	72
	S04	1.9	11.9	36	47	52
	S05	0.5	5.6	37	45	60
	S06	3.4	2.1	32	90	30
	S07	4.5	4.5	41	20	58
	S08	4.1	1.4	45	21	40
	S11	1.5	0.3	36	67	61
	S15	0.7	3.6	38	38	55
	S18	1.5	7.7	44	71	53
	S20	2.6	4.2	34	52	45
	CA21	3.2	4.4	32	52	33
	CA22	3.5	7.1	36	81	53
	CA24	1.6	7	39	54	45
	CA25	1.2	6.5	41	57	51
	S31	2.1	14	32	35	45
Azoo, sperm counts < 0.1 million (*n* = 15)	P1 (S-2017)	3.5	<0.1	40	nd	nd
	P2	4	<0.1	38	nd	nd
	S02	0.5	<0.1	43	50	50
	S09	1.3	<0.1	32	50	50
	S10	0.5	<0.1	30	0	30
	S13	0.9	<0.1	42	0	nd
	S14	<0.1	<0.1	39	0	0
	S16	6.7	<0.1	39	42	30
	S17	0.3	<0.1	35	nd	nd
	S26	1.2	<0.1	36	57	52
	S32	1.5	<0.1	43	nd	nd
	CA23	0.4	<0.1	39	nd	nd
	CA26	0.9	<0.1	36	nd	nd
	TCA26	0.7	<0.1	30	nd	nd
	TS1	0.6	<0.1	34	nd	nd

## Data Availability

The datasets used and analyzed during the current study are available from the corresponding author on reasonable request.

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
