# Peer review of "A Comprehensive Genetic Study of Microtubule-Associated Gene Clusters for Male Infertility in a Taiwanese Cohort"

_ijms, 2023, doi:10.3390/ijms242015363_

Round 1

Reviewer 1 Report

Main problems:

- Microtubule dynamics and associated genes have been shown to be involved in infertility before, and should be discussed in the introduction.

Cerván-Martín M, Bossini-Castillo L, Guzmán-Jiménez A, Rivera-Egea R, Garrido N, Lujan S, Romeu G, Santos-Ribeiro S; IVIRMA Group; Lisbon Clinical Group; Castilla JA, Gonzalvo MDC, Clavero A, Maldonado V, Vicente FJ, Burgos M, Jiménez R, González-Muñoz S, Sánchez-Curbelo J, López-Rodrigo O, Pereira-Caetano I, Marques PI, Carvalho F, Barros A, Bassas L, Seixas S, Gonçalves J, Larriba S, Lopes AM, Palomino-Morales RJ, Carmona FD. Common genetic variation in KATNAL1 non-coding regions is involved in the susceptibility to severe phenotypes of male infertility. Andrology. 2022 Oct;10(7):1339-1350. doi: 10.1111/andr.13221. Epub 2022 Jul 8. PMID: 35752927; PMCID: PMC9546047.

Smith LB, Milne L, Nelson N, Eddie S, Brown P, Atanassova N, O'Bryan MK, O'Donnell L, Rhodes D, Wells S, Napper D, Nolan P, Lalanne Z, Cheeseman M, Peters J. KATNAL1 regulation of sertoli cell microtubule dynamics is essential for spermiogenesis and male fertility. PLoS Genet. 2012;8(5):e1002697. doi: 10.1371/journal.pgen.1002697. Epub 2012 May 24. PMID: 22654668; PMCID: PMC3359976.

Wu X, Yun D, Sang M, Liu J, Zhou L, Shi J, Wang L, Bu T, Li L, Huang Y, Lin D, Sun F, Cheng CY. Defects of microtubule cytoskeletal organization in NOA human testes. Reprod Biol Endocrinol. 2022 Nov 3;20(1):154. doi: 10.1186/s12958-022-01026-w. PMID: 36329464; PMCID: PMC9632130.

- Text for material, methods, results and discussion mixed throughout, hence major rearrangements needed. For example:

Section 2.3 has material which belongs to introduction (109-111 and 118-123) and some to results (113-115). 

Line 179-181: This belongs to discussion, not methods

Line 184-191: This belongs to materials, not results.

Line 197 onwards: these are not results

Line 208-217: these belong to materials and methods

Presentation issues:

Table 1 and Table reproduced poorly in the proof PDF. In table 1, column “Volume” should be title case, like the other columns.

CRTs have nd sperm count volumes. This should be explained in text.

Line 126: 2 -> two

Line 151: use past tense

Line 152: Illumina misspelled.

Line 157-158: Needs more information

Line 166-167: Sanger results not clearly shown anywhere, or the methods details etc. "...performed on an additional group...". Does not really say anything and sounds more like WGS!

Table 3: Instead of numeric region, the allelic variation would be much more useful here

Figure 3: 3D-graphics of columns are used in business, not in biological science. Use 2D and hatched/solid patterns.

Figure 4: The text should be bit more careful regarding the involvement of these genes to the pathway, as there are likely to be others as well. As minimum, it should be mentioned that these are the factors found in this study and are likely to be non-exhaustive.

Line 126: 2 -> two

Line 151: use past tense

Line 152: Illumina misspelled.

Author Response

Please find the detailed responses below

Reviewer 2 Report

General Comments:

The manuscript offers a comprehensive exploration into the genetic mutations associated with male infertility, with a particular emphasis on microtubule-associated genes. While the research's depth and methodology are commendable, there are areas that warrant further attention and clarification.

Specific Comments:

1. Images and Figures:

   - Resolution and Clarity: The images included in the manuscript are of suboptimal resolution. For readers and fellow researchers to derive meaningful insights, it's imperative that all visual data be clear and easily interpretable. 

   - Detailing: The intricate details within the images are not easily distinguishable. This lack of clarity could lead to misinterpretation or oversight of critical data points.

   - Suggestion: It's recommended to revisit the source of these images. If they were captured or generated at a higher resolution initially, ensure that no quality was lost during the manuscript's formatting or submission process. If the source images are of low quality, consider recapturing or regenerating them. Vector-based graphics, especially for diagrams and charts, can be beneficial.

2. Methodology:

   - Sequencing Choices: While the use of both targeted NGS and WES is well-justified, a more in-depth discussion on why these specific methods were chosen over other available sequencing techniques would provide clarity.

   - Patient Selection: The criteria for selecting patients based on sperm concentration is clear. However, elucidating why these specific divisions are crucial for the study's objectives would be beneficial. Are there underlying biological or genetic reasons for these divisions?

3. Results:

   - Gene Findings: The identification of the five affected genes is a pivotal point in the manuscript. Delving deeper into the broader implications of these findings would be enriching. For instance, how do these genes interact with other known genes associated with infertility? Are there potential pathways or mechanisms that these genes influence which could be targeted for treatment?

4. Discussion:

   - Broader Implications: While the evidence supporting the hypothesis is robust, the discussion could benefit from a broader perspective. How might these findings influence current clinical practices, patient counseling, or genetic testing protocols? Additionally, how do these findings fit into the larger landscape of male infertility research?

5. Recommendations:

   - Follow-up Studies: Given the significance of the findings, a natural next step would be to explore potential therapeutic interventions based on the identified genes. A study focusing on potential treatments or interventions would be a valuable addition to the field.

   - Collaboration: Collaborating with experts specializing in genetic imaging or bioinformatics could enhance the quality of the images and data interpretation. Such collaboration could also open doors to new methodologies or insights that might not have been considered.

6. Data Presentation:

   - Tables and Charts: Ensure that all tables, charts, and graphs are labeled correctly, with clear legends and footnotes if necessary. Any abbreviations used should be explained either within the figure/table or in a footnote.

   - Statistical Analysis: While the manuscript mentions the use of statistical tools, a more detailed breakdown of the statistical methods and their results would provide clarity. This includes p-values, confidence intervals, and any other relevant metrics.

7. Conclusion:

   - Future Directions: The conclusion could benefit from a section discussing potential future directions based on the findings. This could include potential clinical applications, further research areas, or implications for genetic counseling.

Comments on the Quality of English Language:

General Observation: The manuscript is generally well-written, but there are areas where the language could be refined for clarity and precision.

Sentence Structure: In some sections, the sentence structure is complex, making it challenging to follow the main idea. Consider breaking longer sentences into shorter, more concise ones to improve readability.

Terminology Consistency: Ensure consistent use of terms throughout the manuscript. For instance, if "male infertility" is used in one section, avoid switching to "infertility in males" in another without a clear reason.

Grammar and Punctuation: There are occasional grammatical errors and punctuation inconsistencies. It would be beneficial to have the manuscript reviewed by a native English speaker or professional editor to rectify these issues.

Use of Abbreviations: While the use of abbreviations is common in scientific writing, it's essential to introduce each abbreviation clearly when it's first used. Also, consider providing a list of abbreviations at the beginning or end of the manuscript for easy reference.

Clarity in Descriptions: In some sections, especially the methodology, the descriptions could benefit from more straightforward language. Avoiding jargon or overly technical terms, unless necessary, can make the manuscript more accessible to a broader audience.

Transitions between Sections: Smooth transitions between sections and paragraphs will enhance the flow of the manuscript. Ensure that each section logically leads to the next, providing readers with a cohesive narrative.

Conclusion Language: The conclusion should be assertive and clear. Avoid using tentative language, and ensure that the main findings are summarized confidently.

Recommendation: Given the importance and potential impact of the research, it would be highly beneficial to invest in professional language editing services. This will not only enhance the manuscript's language quality but also ensure that the research findings are communicated effectively.

Author Response

(The authors gave the same response as above.)
